# Development, Establishment, and Validation of a Model for the Mineralization of Periodontium Remodelling Cells: Cementoblasts

**DOI:** 10.3390/ijms241813829

**Published:** 2023-09-07

**Authors:** Shruti Bhargava, Joachim Jankowski, Erik Merckelbach, Charlotte Elisa Roth, Rogerio Bastos Craveiro, Michael Wolf

**Affiliations:** 1Institute of Molecular Cardiovascular Research, Medical Faculty, RWTH Aachen University, 52062 Aachen, Germany; sbhargava@ukaachen.de (S.B.); emerckelbach@ukaachen.de (E.M.); 2Aachen-Maastricht Institute for Cardiorenal Disease (AMICARE), University Hospital RWTH Aachen, 52062 Aachen, Germany; 3Experimental Vascular Pathology, Cardiovascular Research Institute Maastricht (CARIM), University of Maastricht, 6211 Maastricht, The Netherlands; 4Department of Orthodontics, Dental Clinic, University of Aachen, Pauwelsstr. 30, 52074 Aachen, Germany; croth@ukaachen.de (C.E.R.); rcraveiro@ukaachen.de (R.B.C.); michwolf@ukaache.de (M.W.)

**Keywords:** cementum, diabetic kidney disease, cementoblast, periodontium, calcification

## Abstract

Chronic kidney disease (CKD) patients undergoing dialysis are at high risk of bone fractures. CKD-induced mineral and bone disorder is extended to periodontal disease due to changes in the ionic composition of saliva in CKD patients, dysregulating mineralization, hindering regeneration and thereby promoting the progression of dental complications. Despite the importance of cementum for overall oral health, the mechanisms that regulate its development and regeneration are not well comprehended, and a lack of sufficient in vitro experimental models has hindered research progress. In this study, the impact of experimental conditions on the calcification of cementoblasts was systematically investigated, aimed at establishing a standardized and validated model for the calcification of cementoblasts. The effects of phosphate, calcium, ascorbic acid, β-glycerolphosphate, dexamethasone, and fetal calf serum on the calcification process of cementoblasts were analyzed over a wide range of concentrations and time points by investigating calcium content, cell viability, gene expression and kinase activity. Cementoblasts calcified in a concentration- and time-dependent manner with higher concentrations of supplements cause a higher degree of calcification but decreased cell viability. Phosphate and calcium have a significantly stronger effect on cementoblast calcification processes compared to osteogenic supplements: ascorbic acid, β-glycerolphosphate, and dexamethasone induce calcification over a wide range of osteogenic signalling pathways, with osteopontin being a central target of gene regulation. Conversely, treatment with ascorbic acid, β-glycerolphosphate, and dexamethasone leads to activating only selected pathways, especially promoting bone sialoprotein expression. The developed and validated cementoblast calcification protocol, incubating up to 60% confluent cementoblasts with 1.9 mmol L^−1^ of phosphate supplementation for a reasonable, multi-pathway calcification induction and 10 mmol L^−1^ β-glycerolphosphate, 75 µmol L^−1^ ascorbic acid and 10 nmol L^−1^ dexamethasone for a reasonable osteogenic differentiation-based calcification induction, provides standard in vitro experimental models for better understanding cementoblast function and regeneration.

## 1. Introduction

Chronic kidney disease (CKD) is accompanied by mineral and bone disorder, especially in patients undergoing dialysis, increasing the risk of bone fractures [1]. CKD-mineral and bone disorder (CKD-MBD) is extended to periodontal disease due to changes in the ionic composition of saliva in CKD patients, such as higher phosphate concentrations [2]. Periodontal regeneration is essential for tooth attachment, for recovering from daily sheer stress, orthodontic tooth movement and highly prevalent dental problems such as periodontitis and tooth ankylsosis [3]. Periodontitis is the sixth most common human disease, affecting 11.2% of the global population [3], with the potential to disrupt dental hard tissue and eventual tooth loss. In addition, periodontitis is a well-known comorbidity of prevalent chronic diseases such as chronic kidney disease [4]. Recently, in vivo methods are being developed to study alveolar defects resulting from diseases such as CKD [5], highlighting the need to study the clinical implications of dental dysfunction resulting from other systemic diseases. However, in vitro methods to study the mechanisms involved have not been established yet.

The absence of vessels and cells in the cementum hinders regeneration, which can be overcome by augmenting the function of cementoblasts, as they can migrate to the impaired tooth root, forming the pre-cementum, followed by mineralization [6]. Therefore, a detailed understanding of the mineralization processes of cementoblasts is required to study the regenerative potential of the periodontal tissue-cementum complex for improving wound healing and the quality of regenerative treatment [7]. It is also necessary to understand the pathophysiological changes in this mineralization process in response and comparison to other systemic diseases [7,8].

Both bone formation and vascular calcification are modulated by the comparable osteochondrogenic markers such as alkaline phosphatase, osteopontin, Runt-related transcription factor 2,5′-nucleotidase ecto, matrix gla protein, ectonucleotide pyrophosphatase/phosphodiesterase 1 [9]. However, these mechanisms have not been studied in detail in context of dentoalveolar defects due to a lack of established in vitro models mimicking CKD-like conditions. This hinders the development and establishment of new therapeutic interventions.

Numerous experimental protocols are used to analyze calcification processes. However, a comprehensive systematic approach examining the influence of each component on the calcification of cementoblasts has not been undertaken so far. This hinders the comparison of the outcomes of different studies and impedes data reproducibility through the lack of a standardized protocol.

Therefore, the current study systematically investigated the impact of differential phosphate and calcium concentrations, a combination of differential phosphate and calcium concentrations, differential osteogenic supplements at different incubation times as well as media supplements such as fetal calf serum to optimize a standard protocol for studying calcification in these cells in this study.

This study facilitates the development of an in vitro model to assess new therapeutic interventions for dental disorders accompanying chronic diseases. The adoption of this well-established and validated protocol will contribute to the standardization of experimental procedures when studying calcification in cementoblasts, promoting comparability among diverse research findings.

## 2. Results

### 2.1. Cementoblast Calcification Increases with Increasing Phosphate Concentrations

To investigate the effects of phosphate concentrations on cementoblasts’ calcification, cementoblasts were incubated in the presence of media containing increased phosphate concentrations (0.9–4.3 mmol L^−1^) for 3, 5 or 7 days. Increasing the concentrations of phosphate resulted in a gradual increase in the extent of cementoblast calcification at all different time points up to 12,575.29% ± 1375.56% (*n* = 9; **** *p* ≤ 0.0001) after 7 days (Figure 1A).

### 2.2. Cementoblast Calcification Increases with Increasing Calcium Concentrations

Next, the effects of calcium concentrations on cementoblast calcification were analyzed by incubation of cementoblasts in the presence of culturing media containing increased calcium concentrations (1.8–5.3 mmol L^−1^) for 3, 5 and 7 days, respectively. Increasing the concentrations of calcium ions resulted in a gradual increase in the extent of cementoblast calcification at all different time points up to 2700.98% ± 312.40% (*n* = 9; **** *p* ≤ 0.0001) after 7 days (Figure 1B).

### 2.3. Increasing Phosphate Concentrations in Combination with Increased Calcium Concentrations Augment Cementoblast Calcification

Incubation of cementoblasts with phosphate (3.3 mmol L^−1^) in the presence of 3.8 mmol L^−1^ calcium leads to a significant increase in calcification of the cementoblasts, but also leads to high cell death of up to 82.99% ± 3.25% compared to controls. Reducing the calcium concentration to 2.3 mmol L^−1^ does not significantly increase cell viability (Appendix A). Reduction of Ca^2+^ content to 2 mmol L^−1^, accompanied by a reduction in phosphate concentration gradient, led to the retention of cell viability. Therefore, a fixed low calcium concentration of 2 mmol L^−1^ was used over a range of phosphate concentrations (0.9–3.1 mmol L^−1^) to investigate the calcification-inducing effects of phosphate ions. Increasing the concentration of phosphate resulted in a gradual increase in the extent of cementoblast calcification at all time points of 3, 5 or 7 days, with the highest increase being 6773.40% ± 665.33% after 7 days (*n* = 9; **** *p* ≤ 0.0001) (Figure 1C).

### 2.4. Increasing Calcium Concentrations in Combination with Increased Phosphate Concentration Augment Cementoblast Calcification

Incubation of cementoblasts with 4.3 mmol L^−1^ calcium in the presence of 2.8 mmol L^−1^, 2.3 mmol L^−1^ or phosphate leads to a significant increase in calcification but also leads to high apoptosis with up to 59.16% ± 3.56% cell death. Reducing the phosphate concentration to 2.3 mmol L^−1^ still led to high cell death (Appendix A). Reduction of phosphate content to 1.1 mmol L^−1^, accompanied by a reduction in the calcium concentration gradient, led to the retention of cell viability. Therefore, to investigate the effects of calcium concentrations in the presence of a fixed phosphate concentration of 1.1 mmol L^−1^ on cementoblast calcification, cementoblasts were incubated in the presence of culturing media containing 1.1 mmol L^−1^ phosphate and increasing calcium ion concentrations (1.8–3.9 mmol L^−1^) for 3, 5 and 7 days, respectively. At these conditions, an increased concentration of Ca^2+^ resulted in a gradual increase in the extent of cementoblast calcification when induced for 5 and 7 days, respectively, leading to an increase of up to 4917.41% ± 357.64% (*n* = 9; **** *p* ≤ 0.0001) (Figure 1D).

The conditions that result in stable significant calcification, first, 1.9 mmol L^−1^ of phosphate, second, 2.5 mmol L^−1^ calcium, third, 1.3 mmol L^−1^ phosphate and 0.2 mmol L^−1^ calcium added to DMEM containing 2.5% FCS, were analyzed further via staining and gene expression studies. However, to study the impact of osteogenic supplements, it was imperative to analyze the impact of FCS supplementation on cementoblast calcification.

### 2.5. Cementoblast Calcification Decreases with Increasing Fetal Calf Serum Concentrations

To investigate the effect of fetal calf serum on cementoblast calcification, cementoblasts were incubated in the presence of media containing increasing concentrations of FCS (1–10%) for 7 days. Since FCS is a stimulator of cell growth [10], different FCS concentrations were used to cultivate cementoblasts. A supplementation of 2.5% FCS showed the highest amount of calcium content (1600.94% ± 274.18%) and a significant amount of cell death (39.67% ± 4.95%). With an increase in FCS concentration, cell survival increases from 39.67% ± 4.95% to 134.93% ± 3.95% (*n* = 9; **** *p* ≤ 0.0001) and correspondingly, cementoblast calcification reduces by 71.08% down to 463.00% ± 22.2% (*n* = 9; **** *p* ≤ 0.0001) (Figure 2).

### 2.6. Cementoblast Calcification Is Dependent on Ascorbic Acid Concentrations, Independent of Dexamethasone Supplementation

The impact of ascorbic acid on cementoblast calcification was validated by incubation of cementoblasts in the presence of media containing increasing concentrations of ascorbic acid (25–125 µmol L^−1^) and a fixed β-glycerolphosphate concentration of 10 mmol L^−1^ for 3, 5 or 7 days in the absence or presence of 10 nmol L^−1^ dexamethasone. The absence or presence of dexamethasone did not show a significant difference in cementoblast calcification (Figure 3A,B). A 3 day induction of calcification does not show a significant increase in calcium content as compared to the control medium (Figure 3A,B). With an increase in ascorbic concentration, there was a significant increase in cementoblast calcification. An increased ascorbic concentration caused a significant increase in cementoblast calcification of 1561.36% ± 240.33% (*n* = 9, **** *p* ≤ 0.0001).

### 2.7. Cementoblast Calcification Is Dependent on β-Glycerolphosphate Concentrations

To investigate the effect of β-glycerolphosphate concentrations on cementoblast calcification, cementoblasts were incubated in the presence of media containing increasing concentrations of β-glycerolphosphate (0–10 mmol L^−1^), in the presence or absence of 75 µmol L^−1^ ascorbic acid for 3, 5 and 7 days. The absence or presence of 75 µmol L^−1^ ascorbic acid did not show a significant difference in cementoblast calcification (Figure 3C,D). A 3 day induction of calcification does not show a significant amount of calcification as compared to the control medium (Figure 3C,D). With an increase in β-glycerolphosphate concentration, there was a significant increase in cementoblast calcification when induced for 5 or 7 days, with 7 days of induction leading to the highest calcium concentrations, showing an increase of up to 469.67% ± 103.38% (*n* = 9, **** *p* ≤ 0.0001) (Figure 3).

### 2.8. Visualization and Quantification of the Calcification of Cementoblasts by Von Kossa Staining and Alizarin Red Staining

Both von Kossa and alizarin red staining are widely used methods for staining calcium deposition post-culture termination. Therefore, both methods were used to visualize Ca^2+^ deposition in cementoblasts. Three conditions from calcium and phosphate combinations were analyzed to identify the formation of intracellular or extracellular mineralized nodules: (a) the calcification-inducing condition was 1.9 mmol L^−1^ of phosphate, (b) 2.5 mmol L^−1^ calcium and (c) 1.3 mmol L^−1^ phosphate and 0.2 mmol L^−1^ Ca^2+^ added to DMEM containing 2.5% FCS. In addition, three conditions from osteogenic supplement combinations: (a) 75 µmol L^−1^ ascorbic acid, 10 mmol L^−1^ β-glycerolphosphate, and 10 nmol L^−1^ dexamethasone; (b) 75 µmol L^−1^ ascorbic acid, 12.5 mmol L^−1^ β-glycerolphosphate, and 5 nmol L^−1^ dexamethasone and (c) 75 µmol L^−1^ ascorbic acid, 7.5 mmol L^−1^ β-glycerolphosphate, and 5 nmol L^−1^ dexamethasone, each incubated for 7 days at 37 °C, were further analyzed. Therefore, six different calcification-inducing conditions (Table 1) were analyzed by von Kossa staining (Figure 4). When quantified using the ’Image J software’ (V1.52), each analyzed condition confirmed the calcium measurements of a significant increase in the degree of cementoblast calcification (Figure 4B,D).

The same six calcification-inducing conditions were stained with alizarin red to identify the calcification-inducing conditions most suited for analysis across different calcification assays (Figure 5). The quantification of calcification using the ’Image J’ software V1.52 shows a significant increase in cementoblast calcification as expected based on calcium measurements (Figure 5B,D). The quantification of alizarin red staining was 53.27% ± 1.88% as compared to von Kossa staining for media using phosphate or calcium as supplements. Media containing osteogenic supplements led to alizarin red staining of only 37.03% ± 9.66% compared to von Kossa staining, demonstrating that von Kossa stained more area from the same conditions compared to alizarin red staining for cementoblasts.

### 2.9. Different Calcification Pathways Are Triggered by Different Calcification-Inducing Supplements

The mRNA level of osteogenic genes such as osteopontin and bone sialoprotein was quantified in the six calcifying conditions, which showed the most consistent induction of calcification evaluated by both calcium content quantification and staining to clarify which pathways are triggered by different calcification-inducing supplements. Supplementation with calcium and phosphate significantly increased osteopontin expression after 12 h of calcification induction (Figure 6A). However, osteogenic supplements did not significantly induce osteopontin expression after 12 or 24 h (Figure 6A). A significant increase in bone sialoprotein expression was observed in cells supplemented with β-glycerolphosphate, ascorbic acid and dexamethasone-induced for 24 h (Figure 6B). However, supplementation with calcium and phosphate did not significantly induce the expression of bone sialoprotein at either time point (Figure 6B).

### 2.10. Kinases Influencing Osteogenic Pathways Are Phosphorylated by the Calcification-Inducing Supplements

Kinome analyses were performed to characterize the genomic basis of osteogenic potential of specific calcification-inducing supplements. The mitogen-activated protein kinases (MAPK) represent a non-canonical pathway, which induces osteogenic gene expression, influencing calcification [11]. In line with the osteopontin expression, kinome analysis revealed that 48 h after supplementation with 1.9 mmol L^−1^ of phosphate led to a substantial increase in extracellular signal-regulated kinases (ERK), p38 and c-Jun N-terminal kinases (JNK) activity, together with an increase in EphB4-B2 pathway, Janus kinase (JAK) and Rho-associated coiled-coil kinases (ROCKs) activation (Figure 7A,B: Blue), compared to cell culture in control conditions. Conversely, CDK9, IKK and 5′-AMP-activated protein kinase activation were strongly upregulated in the presence of osteogenic supplements 12.5 mmol L^−1^ β-glycerolphosphate, 75 µmol L^−1^ ascorbic acid and 5 nmol L^−1^ dexamethasone after induction of 48 h, compared to cell culture in control conditions (Figure 7A,B: Green).

## 3. Discussion

Systemic diseases such as chronic kidney disease, cardiovascular disease or diabetes are accompanied by comorbidities of periodontal tissue degeneration [3,12]. Periodontal disease is a degenerative disease of the tooth supporting apparatus and often accompanied with a chronic periodontal tissue inflammation triggered by the host immune response to pathogenic bacteria [3,13,14]. However, demineralization is reversible through exposure to oral environments that favor demineralization, promoting the growth of partially demineralized hydroxyapatite crystals in teeth to their original size [15].

While in healthy subjects, the circulatory phosphate levels range between 0.81 and 1.45 mmol L^−1^, and circulatory calcium levels range between 2.1 and 2.6 mmol L^−1^ [16], under pathophysiological conditions, phosphate and calcium plasma levels are elevated up to 2.85 mmol L^−1^ and 2.95 mmol L^−1^, respectively [16], potentially forming the basis for imbalanced mineralization [10]. Renal disease or dental decay show a positive correlation between serum and saliva phosphate levels. Under disease conditions, saliva phosphate concentrations may go up to 13.7 ± 4.4 mg/dL which corresponds to 4.4 ± 1.4 mM [17].

Of the three dental calcified hard tissues, cementum, dentine and enamel, cementum and bone consist of similar amounts of the organic matrix, which consists of up to 90% of collagen type I in both tissues and hydroxyapatite. The cementum exhibits highly regenerative capabilities and is often regarded to be an important tissue in periodontal regeneration [18]. Markers such as osteocalcin distinguish cementoblasts, the cellular component of cementum, from other dental cells, such as periodontal ligament fibroblasts [19]. Elevating phosphate, calcium, and combinations are widely used to induce calcification [16,20]. Additionally, β-glycerolphosphate, ascorbic acid and dexamethasone are osteogenic and are known to induce calcification [21]. As such, furthering our understanding of cementum mineralization is imperative in the treatment of tooth loss and demineralization of dental hard tissue and for the innovation and development of scaffolds.

Therefore, a systematic approach to study the influence of each factor influencing calcification was undertaken with regard to cementoblasts. In the present study, we demonstrate that phosphate and calcium ions at different concentrations, their combinations, and combinations of β-glycerolphosphate, ascorbic acid and/or dexamethasone induce varying degrees of calcification in cementoblasts. Due to the strong correlation between calcification and cell death, identifying the combination of supplements for a minimum adequate time, that do not trigger calcification through apoptosis is a necessity [22]. The addition of phosphate to the calcification-inducing medium led to exponentially higher calcium concentrations as compared to the addition of calcium when the cells were incubated for up to 7 days. A combination of high concentrations of phosphate and calcium ions led to more pronounced calcification but also significant cell death, indicating that phosphate concentrations above 3.3 mmol L^−1^ with 2.3 mmol L^−1^ calcium and calcium concentration above 4.3 mmol L^−1^ with 2.3 mmol L^−1^ phosphate induce significant calcification that is correlated to apoptosis. Therefore, to ensure significant cell survival with active calcification, a slight increase of either calcium (from 1.8 inherently present in DMEM to 2 mmol L^−1^) or phosphate (from 0.9 inherently present in DMEM to 1.1 mmol L^−1^) in combination with gradients of phosphate and calcium ions is suitable.

In addition to the phosphate or calcium ion dose- and time-dependency of calcification processes, we studied the impact of FCS concentration on cementoblast calcification. The FCS was heat inactivated to decrease the titer of heat labile complement proteins [23]. Lower concentrations of FCS, e.g., less than 1%, have been reported to be cytotoxic and can lead to higher apoptosis [24]. Supplementation above 2.5% FCS promotes cementoblast survival, coinciding with a steady dose-dependent decrease in calcification. An increase in FCS concentration leads to a decrease in calcium content due to a high cell survival rate, which hinders the formation of nucleation sites in the form of apoptotic cells, which promote calcium accumulation. FCS is known to contain calcification inhibitors such as matrix Gla proteins (MGP) [25] and fetuin-A [26], an inflammation-related Ca-regulatory glycoprotein with multiple Ca-binding sites [27].

β-glycerolphosphate, ascorbic acid and dexamethasone are traditional supplements used for the osteogenic differentiation of mesenchymal stem cells and, therefore, are used to induce calcification [28]. Induction of calcification by β-glycerolphosphate, ascorbic acid and dexamethasone supplementation requires a minimum incubation time of five days, which is potentially caused due to limited activation of osteogenic pathways by these supplements.

β-glycerolphosphate is a relevant component for inducing calcification as it leads to the expression of osteogenic genes and causes mitochondrial oxidative stress, a factor known to promote calcification [29]. Calcification induced by increasing calcium and phosphate combinations was observed to be much more pronounced compared to β-glycerolphosphate, ascorbic acid and dexamethasone-induced calcification at all time points, indicating that cementoblasts calcify slowly in response to these osteogenic supplements. However, the presence or absence of dexamethasone did not impact the calcium content of the cementoblasts. An incubation time of 7 days is optimal to induce stable calcification of cementoblasts without leading to significant cell death.

Ascorbic acid is an essential nutrient supporting extracellular matrix production by mediating the activation of mammalian target of rapamycin (mTOR) pathway [30,31], and it protects against rapid calcification for up to 5 days of incubation with β-glycerolphosphate, which was overcome at longer incubation times. A minimum incubation of 7 days was required to induce significant calcification when inducing calcification with ascorbic acid in addition to β-glycerolphosphate as a component of the calcification medium.

Calcium measurements were confirmed via von Kossa and alizarin red staining, showing in all calcifying media a significant increase in calcification compared to controls. While von Kossa is non-specific for calcium, alizarin red S reacts with calcium cation to form a chelate, facilitating the visualization of calcified areas [32]. Von Kossa staining of cementoblasts led to a more stained area for the same conditions than alizarin red staining, which is more specific for calcium. This highlights the mechanistic difference between the two different calcification-inducing methods, as the calcium-specific alizarin red staining of cementoblasts could stain half the area stained by von Kossa staining when calcium or phosphate was used as a supplement. However, the calcium-specific alizarin red staining of cementoblasts could stain only one-third of the area stained by von Kossa staining when osteogenic supplements were used, implying that calcium- and phosphate-induced calcification is more calcium dense.

Additionally, it is important to note that the measurement of calcium content was normalized to the protein content of the cells, while staining was calculated as a percentage of the area stained by the dyes. The degree of staining positively correlates with the degree of calcification, i.e., the amount of incorporated calcium. However, staining is best employed as a visualization of calcification, and calcium measurement should be used to quantify the precise cellular calcium concentration, as shown in Figure 1, Figure 2 and Figure 3.

The quantification of osteogenic genes 12 h after calcification induction by high calcium ions or phosphate shows a marked increase in osteopontin, which plays an essential role in deciding the osteogenic fate of mesenchymal stem cells [33]. The restoration of osteopontin levels to baseline at later time points demonstrates that calcification via high phosphate and calcium ion concentrations is based on a fast induction of osteocalcin. Conversely, calcification induction by β-glycerolphosphate, ascorbic acid and dexamethasone is mediated through increased expressions of bone sialoprotein, which is expressed 24 h after the induction of calcification, demonstrating that these supplements trigger calcification through osteogenic differentiation.

As a further validation of the osteogenic gene expression, induction of cementoblast calcification via phosphate supplementation was mediated by significant upregulation in the activity of osteopontin-associated serine/threonine and tyrosine kinases. The majority of the identified serine/threonine kinases belong to the MAPK pathway, which regulates osteopontin expression via the downstream activation of MAPKK, p38, ERK 1, 2, and 5, and JNK [34,35,36] (Figure 7). Furthermore, the protein kinase A, G, and C (AGC) superfamily’s rho-associated coiled-coil kinases (ROCKs) 1 and 2 were also upregulated and play a significant role in regulating osteogenic differentiation and the expression of osteogenic genes like osteopontin [37]. The EphB4-B2 pathway, crucial in regulating bone formation, osteogenic cell differentiation, and osteopontin expression [38] was upregulated in cementoblasts treated with increased phosphate concentration. Since calcification and apoptosis often go hand in hand, it is important to consider that p38 [39], JNK [39], ERK [40] and ROCK [41] activation promotes apoptosis. Conversely, no upregulation of the mentioned kinases was observed upon induction with β-glycerolphosphate, ascorbic acid, and dexamethasone, indicating that these compounds affect different pathways at different time- points. Cyclin-dependent kinases control osteoblast differentiation in addition to regulating the cell cycle [42]. In addition, the cyclin-dependent kinase CDK9 was upregulated upon treatment with these osteogenic supplements, promoting inflammation by activating the NF-κB pathway [43]. The presence of osteogenic supplements caused an upregulation of kinase IKKα, which phosphorylates the NF-κB inhibitor IκBα, leading to ubiquitin-dependent IκBα degradation and subsequent NF-κB activation [44]. 5′ AMP-activated protein kinase activation in the presence of osteogenic supplements also correlates to the increased expression of bone sialoprotein, a gene downstream of 5′ AMP-activated protein kinase pathway [45]. These results indicate that treatment with phosphate leads to a quick upregulation and activation of various osteogenic pathways, while treatment with β-glycerolphosphate, ascorbic acid, and dexamethasone leads to a slow and limited activation of osteogenic pathways.

Based on all these parameters and aiming for one protocol for inducing calcification of cementoblasts, a calcification protocol was standardized and evaluated to study cementoblast calcification. In this protocol, cementoblast calcification was induced by incubating 60% confluent cementoblasts with 1.9 mmol L^−1^ of phosphate supplementation for a quick, multi-pathway calcification induction and 10 mmol L^−1^ β-glycerolphosphate, 75 µmol L^−1^ ascorbic acid and 10 nmol L^−1^ dexamethasone for a slow osteogenic differentiation-based calcification induction. As a control medium, DMEM with the final phosphate and calcium ion concentrations of 0.9 mmol L^−1^ and 1.8 mmol L^−1^, respectively, was used. Calcification analysis in cementoblasts was performed by calcium measurement using the o-cresolphthalein complexone method.

In conclusion, in the current study, we developed and validated a standardized operating protocol (SOP) for systematic calcification of cementoblasts, to study the mineralization processes of the periodontal tissue-cementum complex and for understanding the pathophysiological changes in this mineralization process in response to and comparison with other systemic diseases. Application of this in vitro protocol can facilitate studies focused on therapeutic interventions for clinical practice. This would lead to improved diagnostics and treatment strategies for dental conditions related to chronic diseases.

Limitations: This study demonstrates that different calcification-inducing supplements trigger different calcification-inducing mechanisms. This mandates prior in-depth knowledge and decision about the research question and expected affected mechanisms. If no such decision is made beforehand, two different protocols need to be followed to study the effect of mediators of dental calcification.

Moreover, this study is based on supplementation of cementoblasts with physiologic concentrations of calcification inducers; however, as an in vitro study, it is limited in its inability to fully replicate the complex interactions and physiological conditions of living organisms. This may lead to results that do not accurately reflect in vivo conditions.

## 4. Materials and Methods

### 4.1. Cell Culture

Immortalized murine osteocalcin expressing cementoblasts (OC/CM), kindly provided by Martha J. Somerman [46,47,48], were cultured in DMEM low glucose (1 g L^−1^) (Gibco, Billings, MT, USA, 31885049) 10% FCS (Gibco, Billings, MT, USA, 10500064), 100 units mL^−1^ of penicillin and 100 g mL^−1^ of streptomycin (Gibco, Billings, MT, USA, 15240-062) in cell culture plates and incubated at 37 °C and 5% CO_2_ in a humidified atmosphere. Cells were trypsinized, centrifuged at 350 g, and quantified using a Neubauer Counting Chamber. Approx. 1 × 10^4^ cells were plated in each 48-well for calcification experiments, and 5 × 10^4^ cells were plated in each 12-well for staining and gene expression experiments. Sterile filtrated bovine serum was acquired from Gibco. Additionally, before usage, the FCS was heat inactivated for 30 min at 56 °C in a water bath and filtered again under sterile conditions. The FCS was divided into aliquots and stored at −20 °C before use.

### 4.2. Basic Incubation Medium of Cementoblasts

High-glucose Dulbecco’s Modified Eagle’s Medium (DMEM), already containing 0.9 mmol L^−1^ phosphate, 1.8 mmol L^−1^ calcium and 1% penicillin-streptomycin, was used as the reference control medium.

### 4.3. Inducers of Calcification Processes in Cementoblasts for Experiments

To investigate the effect of phosphate and calcium concentrations on the calcification of cementoblasts, increasing concentrations of Na_2_HPO_4_ and NaH_2_PO_4_ (0.9–4.3 mmol L^−1^) and/or CaCl_2_ (2.3–5.3 mmol L^−1^) were added to the DMEM with 1% P/S and 2.5 to 10% FCS just before use. Different concentrations of β-glycerolphosphate (1.5–10 mmol L^−1^), ascorbic acid (25–125 µmol L^−1^) and dexamethasone (10 nmol L^−1^) were added to DMEM with 1% P/S and 2.5 to 10% FCS before use. The composition of the media used for each condition is summarized in (Table 2).

### 4.4. Quantification of the Calcium Concentration of Cementoblasts

For calcium content quantification, cementoblasts incubated with the respective treatments for 3, 5 or 7 days at 37 °C, 5% CO_2_ level in a humidified atmosphere with 95% relative humidity, were washed 3 times with phosphate buffered saline (PBS) (calcium chloride and magnesium chloride-free PBS (Sigma-Aldrich, Darmstadt, Germany)). The cells were decalcified by treatment with 120 µL of 0.1 mmol L^−1^ HCl solution for 1 h at 4 °C by gentle shaking on a plate shaker. RANDOX Ca^2+^ detection kit (Randox Laboratories, Crumlin, UK) was used according to the manufacturer’s protocol to quantify the calcium content of cementoblasts under different conditions. A linear regression standard curve was generated using a serial dilution series of known calcium concentrations provided by the kit for standardizing the measurements. UV absorption was measured for each well at a wavelength of 570 nm using a Tecan Plate Reader Infinite m200 (Tecan group AG, Männerdorf, Switzerland). The calcium content was normalized to the total protein amount of the cells as described in Section 4.5.

### 4.5. Quantification of Protein Concentration of Cementoblasts

The total protein concentration was quantified from each well to determine the calcium accumulation with reference to cell viability. The acidic environment of the cells was neutralized by treatment with 100 µL 0.1 mol L^−1^ NaOH-sodium dodecyl sulfate (SDS) containing 0.2% SDS and 200 µL phosphate-buffered saline (PBS). The cells were lysed, denaturing the proteins by incubating the plate for 1 h at room temperature. The protein concentration was quantified colorimetrically by the micro bicinchoninic acid (BCA) protein assay kit (Thermo Scientific, Waltham, MA, USA) according to the manufacturer’s protocol. A regression standard curve was generated using a serial dilution series of known bovine serum albumin (BSA) concentrations provided by the kit starting from 40 µg mL^−1^ up to 0.625 µg mL^−1^ for standardizing the measurements of each well. A Tecan Plate Reader Infinite m200 (Tecan group AG, Männerdorf, Switzerland) microplate spectrophotometer was used to measure the absorption at 562 nm. The calcium content was normalized to protein content in each well to indicate the final calcium concentrations.

### 4.6. Visualization and Quantification of the Calcification of Cementoblasts by Von Kossa Staining

Von Kossa staining was performed to visualize the calcification of cementoblasts in six calcifying conditions, which showed the most consistent induction of calcification. Incubation media was removed from the 12-well plate. In detail, each well was washed with 1 mL PBS three times. The cementoblasts were fixed by incubating at 4 °C for 1 h with 300 µL 4% PFA. The cells were incubated with 500 µL 5% aqueous silver nitrate solution in 405 nm UV light for 15 min. The cells were washed three times with double distilled water and incubated in the presence of 500 µL 5% aqueous sodium thiosulfate for 1 min and washed three times with double distilled water. The cells were incubated with a nuclear fast red solution at room temperature for 5 min and rinsed with double distilled water. The cells were covered with a clear mounting solution Vitro-Clud (R. Langenbrinck GmbH, Emmendingen, Germany). The stained cementoblasts were visualized and photographed using a Leica microscope (Leica, Wetzlar, Germany). The calcified area in each well was quantified using the publicly available Java-based image processing software “Image J” (V1.52) [49].

### 4.7. Visualization and Quantification of the Calcification of Cementoblasts by Alizarin Red Staining

Alizarin red staining was performed to visualize the calcification of cementoblasts in 6 calcifying conditions showing the most consistent induction of calcification. Incubation media was removed from the 12-well plate. In detail, each well was washed with PBS three times. The cementoblasts were fixed by incubating them at 4 °C for 1 h with 300 µL 4% PFA. The cells were incubated with 2% Alizarin red S dye (Sigma-Aldrich, Darmstadt, Germany) for 1 h in the dark and then washed with double distilled water until the supernatant was clear to remove any remnant artefacts from the dye.

### 4.8. Reverse Transcription-Quantitative Polymerase Chain Reaction (RT-qPCR) Analyses of mRNA Expression

According to the manufacturer’s instructions, the total RNA was extracted from cementoblasts using an ‘RNAeasy mini kit’ (Qiagen, Hilden, Germany). A quantity of 1 µg total RNA, random hexamers and ‘Verso reverse transcriptase’ (Thermo Fisher Scientific, Waltham, MA, USA) were used to perform reverse transcription as per the manufacturer’s instructions. The ‘LightCycler 480 system’ (Roche Applied Sciences, Penzberg, Germany) was used to determine the gene expression levels by real-time PCR using SYBR Green I dye chemistry (Thermo Fisher Scientific, Waltham, MA, USA). The PCR primers (Table 3) were designed using NCBI primer blast and purchased via Eurofins Genomics. Gene expression levels were quantified relative to the standard curves with the LightCycler analysis software (version 3.5). Data were represented as the mean of gene expression relative to the gene expression of the reference gene.

### 4.9. Kinase Assay

Kinase activity profiling was carried out using PamChip^®^ serine/threonine (Ser/Thr) Kinase assay (STK; PamGene International, ‘s-Hertogenbosch, The Netherlands). The STK-PamChip^®^ array consists of 144 unique phospho-sites that are peptide sequences derived from substrates for Ser/Thr kinases. Three biological replicates per condition of cementoblasts were used for the Assay. After the respective treatments, the cells were washed with ice-cold PBS and lysed using M-PER Mammalian Extraction Buffer containing ‘Halt Phosphatase Inhibitor’ and EDTA-free ‘Halt Protease Inhibitor Cocktail’ (1:10 each; Sigma-Aldrich, Darmstadt, Germany) for 15 min on ice. The lysates were centrifuged at 16,000× *g* for 15 min at 4 °C in a pre-cooled centrifuge. Protein quantification was performed with the Pierce™ Coomassie Plus (Bradford, UK) Assay according to the manufacturer’s instructions. For the assay, 0.5 µg of protein and 400 µmol L^−1^ ATP were applied per array (*n* = 3 per condition) along with an antibody mixture to detect the phosphorylated kinase. The sample was incubated at 30 °C for 1 h while being pumped back and forth through the porous material to enhance binding kinetics and minimize assay time. A second FITC-conjugated antibody was used to detect the phosphorylation signal, and imaging was performed using an LED imaging system. The spot intensity at each time point was quantified using the BioNavigator software version 6.3 (PamGene International, ‘s-Hertogenbosch, The Netherlands). Upstream kinase analysis [12] was utilized as the functional scoring method to rank kinases based on the combined specificity scores from peptides linked to a kinase acquired from six databases. Sensitivity scores were determined by calculating the treatment–control differences.

### 4.10. Statistics

GraphPad Prism Ver. 9 software (GraphPad Software, San Diego, CA, USA) was used for the statistical analysis, and the data were represented as mean ± SEM. Analysis of variance (26) was used to determine the differences between calcification-inducing groups and the reference group for a single factor (one-way) and multiple factors (two-way). Bonferroni’s multiple comparisons were used as a post-test. Differences at * *p* < 0.05, ** *p* ≤ 0.01, *** *p* ≤ 0.001, **** *p* ≤ 0.0001 were considered to be statistically significant.

## Figures and Tables

**Figure 1 ijms-24-13829-f001:**
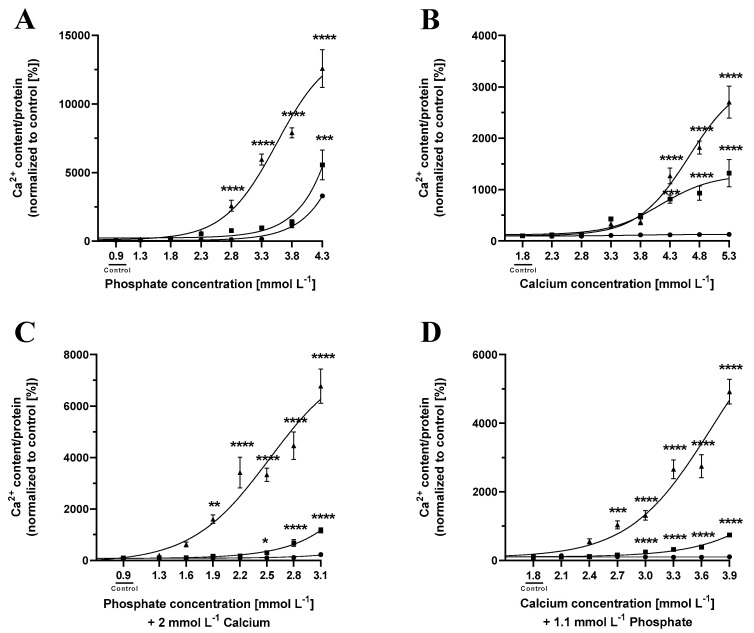
Cementoblasts calcify in a concentration and time-dependent manner in response to phosphate or calcium. Cementoblasts were incubated for 3 (circle), 5 (square) or 7 (triangle) days in the presence of increasing phosphate concentrations (**A**) or increasing calcium ion concentrations (**B**), increasing phosphate concentrations with an elevated calcium concentration of 2 mmol L^−1^ (**C**), increasing calcium concentrations with elevated phosphate concentration of 1.1 mmol L^−1^ (**D**). Data are shown as means ± SEM (*n* = 9). * *p* < 0.05, ** *p* ≤ 0.01, *** *p* ≤ 0.001 and **** *p* ≤ 0.0001 compared to the samples with the same supplement concentrations after 3 days of incubation based on one-way ANOVA. Bonferroni’s multiple comparisons were used as a post-test.

**Figure 2 ijms-24-13829-f002:**
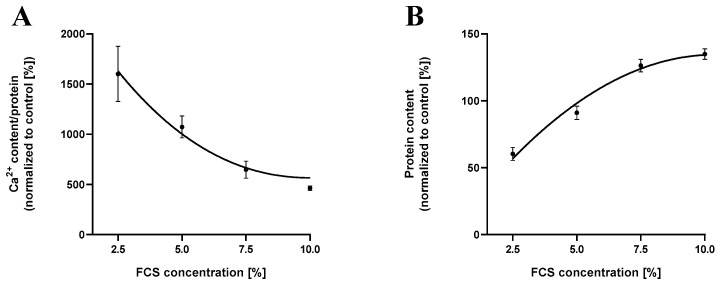
The extent of cementoblast calcification is dependent on FCS supplementation. Cementoblast mineralization decreases with an increase in FCS supplementation up to 10% (**A**), which is in line with an increase in cell survival (**B**). Cementoblasts were incubated for 7 days in the presence of increasing FCS concentrations. Data are shown as means ± SEM (*n* = 9).

**Figure 3 ijms-24-13829-f003:**
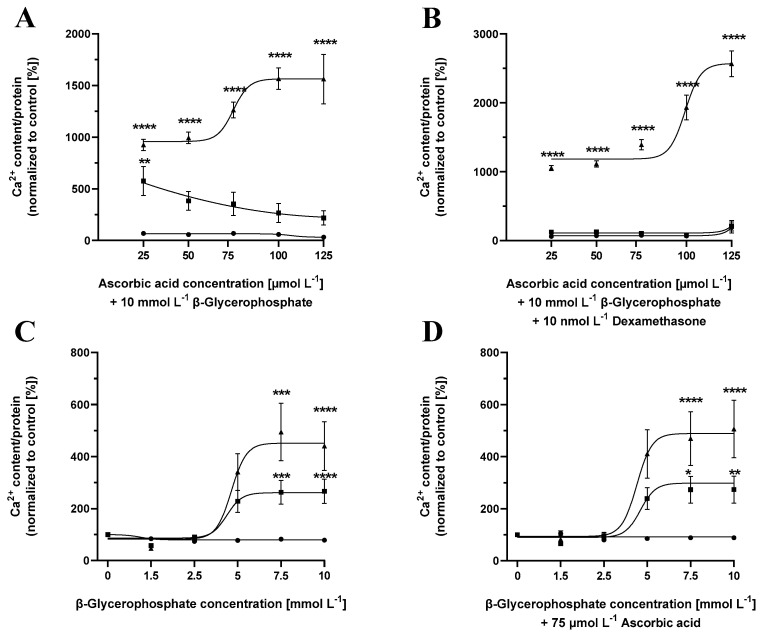
Cementoblasts calcify after 7 days of incubation with ascorbic acid or β-glycerolphosphate supplementation. Cementoblasts were incubated for 3 (circle), 5 (square) or 7 (triangle) days in the presence of 10 mmol L^−1^ β-glycerolphosphate and increasing ascorbic acid concentrations in the presence (**A**) and absence (**B**) of 10 nmol L^−1^ dexamethasone, or a range of β-glycerolphosphate concentrations in the presence (**C**) or absence (**D**) of 75 µmol L^−1^ ascorbic acid. Data are shown as means ± SEM (*n* = 9). * *p* < 0.05, ** *p* ≤ 0.01, *** *p* ≤ 0.001 and **** *p* ≤ 0.0001 compared to the samples with the same supplement concentrations after 3 days of incubation based on one-way ANOVA. Bonferroni’s multiple comparisons were used as a post-test.

**Figure 4 ijms-24-13829-f004:**
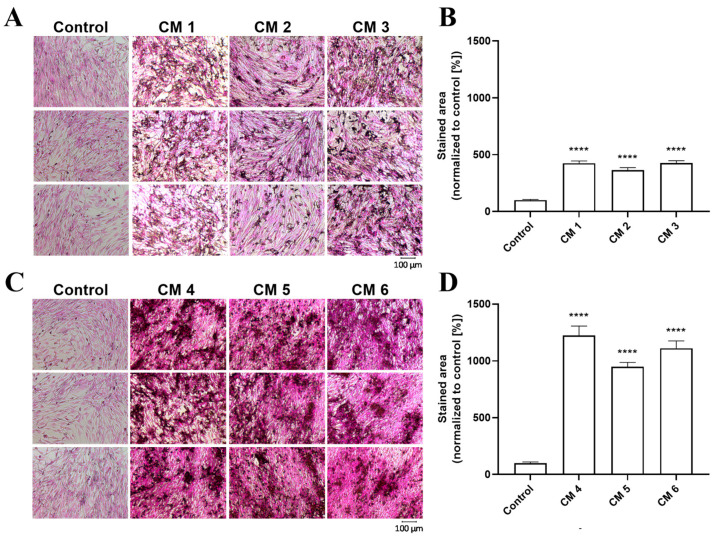
Visualization and quantification of cementoblast calcification via von Kossa staining. Cementoblasts were incubated for 7 days with 6 different calcification conditions: the first, 1.9 mmol L^−1^ of phosphate, the second, 2.5 mmol L^−1^ calcium, the third, 1.3 mmol L^−1^ phosphate and 0.2 mmol L^−1^ calcium added to DMEM containing 2.5% FCS. The fourth condition was 75 µmol L^−1^ ascorbic acid, 10 mmol L^−1^ β-glycerolphosphate, and 10 nmol L^−1^ dexamethasone, the fifth, 75 µmol L^−1^ ascorbic acid, 12.5 mmol L^−1^ β-glycerolphosphate, and 5 nmol L^−1^ dexamethasone and the sixth, 75 µmol L^−1^ ascorbic acid, 7.5 mmol L^−1^ β-glycerolphosphate, and 5 nmol L^−1^ dexamethasone added to DMEM containing 5% FCS. The cells were stained by von Kossa staining and imaged. Shown are the representative images of von Kossa staining from conditions 1–3 (**A**) and conditions 4–6 (**C**), and the quantification of the percentage of stained area of condition 1–3 (**B**) and condition 4–6 (**D**). Data are shown as means ± SEM (*n* = 9). **** *p* ≤ 0.0001 compared with the control based on one-way ANOVA. Bonferroni’s multiple comparisons were used as a post-test.

**Figure 5 ijms-24-13829-f005:**
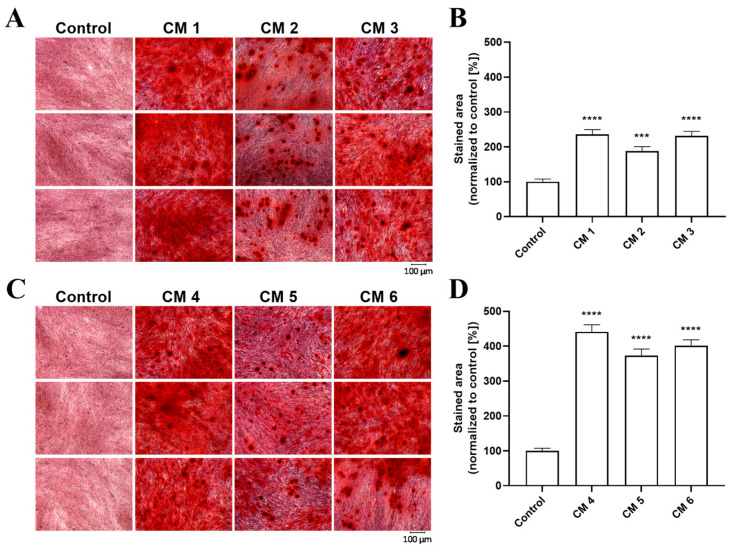
Visualization and quantification of cementoblast calcification via alizarin red staining. Cementoblasts were incubated for 7 days with 6 different calcification conditions: the first, 1.9 mmol L^−1^ of phosphate, the second, 2.5 mmol L^−1^ calcium, the third, 1.3 mmol L^−1^ phosphate and 0.2 mmol L^−1^ calcium added to DMEM containing 2.5% FCS. The fourth condition was 75 µmol L^−1^ ascorbic acid, 10 mmol L^−1^ β-glycerolphosphate, and 10 nmol L^−1^ dexamethasone, the fifth, 75 µmol L^−1^ ascorbic acid, 12.5 mmol L^−1^ β-glycerolphosphate, and 5 nmol L^−1^ dexamethasone and the sixth, 75 µmol L^−1^ ascorbic acid, 7.5 mmol L^−1^ β-glycerolphosphate, and 5 nmol L^−1^ dexamethasone added to DMEM containing 5% FCS. The cells were stained by alizarin red staining and imaged. Shown are the representative images of alizarin red staining from conditions 1–3 (**A**) and conditions 4–6 (**C**), and the quantification of the percentage of stained area of condition 1–3 (**B**) and condition 4–6 (**D**). Data are shown as means ± SEM (*n* = 9). *** *p* ≤ 0.001 and **** *p* ≤ 0.0001 compared with the control based on one-way ANOVA. Bonferroni’s multiple comparisons were used as a post-test.

**Figure 6 ijms-24-13829-f006:**
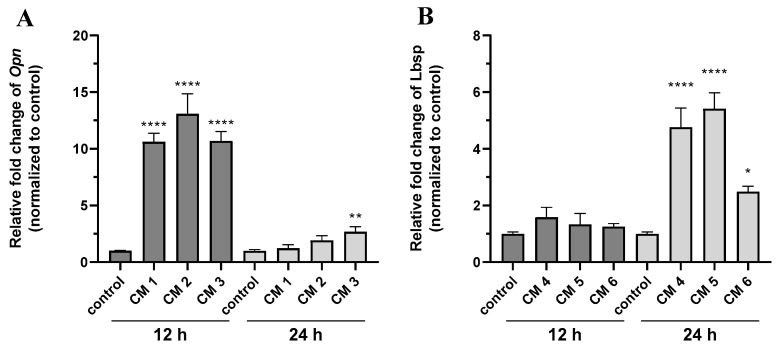
Phosphate and calcium supplementation promote *Opn* expression. The combination of ascorbic acid, β-glycerolphosphate and dexamethasone promote *Lbsp* expression. Cementoblast were incubated for 12 (dark grey) or 24 (light grey) h, respectively, with 6 different calcification conditions: the first, 1.9 mmol L^−1^ of phosphate, the second, 2.5 mmol L^−1^ calcium, the third, 1.3 mmol L^−1^ phosphate and 0.2 mmol L^−1^ calcium added to DMEM containing 2.5% FCS. The fourth condition was 75 µmol L^−1^ ascorbic acid, 10 mmol L^−1^ β-glycerolphosphate, and 10 nmol L^−1^ dexamethasone, the fifth condition was 75 µmol L^−1^ ascorbic acid, 12.5 mmol L^−1^ β-glycerolphosphate, and 5 nmol L^−1^ dexamethasone and the sixth condition was 75 µmol L^−1^ ascorbic acid, 7.5 mmol L^−1^ β-glycerolphosphate, and 5 nmol L^−1^ dexamethasone added to DMEM containing 5% FCS. Relative quantification of (**A**) *Opn* and (**B**) *Lbsp* expression. Data are shown as means ± SEM (*n* = 9). * *p* < 0.05, ** *p* ≤ 0.01, and **** *p* ≤ 0.0001 compared with the control based on one-way ANOVA. Bonferroni’s multiple comparisons were used as a post-test.

**Figure 7 ijms-24-13829-f007:**
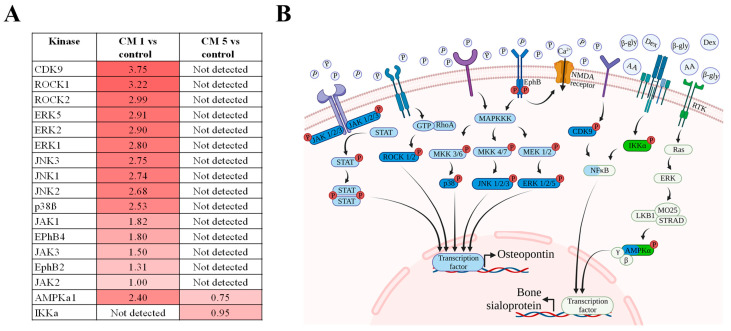
Overview of kinases regulating the effect of calcifying media on cementoblasts. Shown are (**A**) a table of kinases with upregulated activity in calcifying media as seen in the PamChip^®^ assay, and (**B**) an overview of signalling pathways mediating calcification in cementoblasts. Shown are the kinases mediating phosphate-induced calcification (blue) and the calcification induced by ascorbic acid, β-glycerolphosphate, and dexamethasone (green). The kinases marked in dark blue or green show elevated levels of activity in calcifying media, as seen per PamChip^®^ assay. JAK-janus kinase; STAT-signal transducers and activators of transcription; GTP-guanosine-5′-triphosphate; RhoA-ras homolog family member A; ROCK-rho-associated, coiled-coil-containing protein kinase; MAPKKK-mitogen activated protein kinase kinase kinase; MKK-mitogen-activated protein kinase kinase; JNK-c-Jun N-terminal kinases; ERK-extracellular signal-regulated kinases; NMDA-N-methyl-D-aspartate receptor; CDK-cyclin-dependent kinase; NFκB-nuclear factor ‘kappa-light-chain-enhancer’ of activated B-cells; RTK-receptor tyrosine kinase; LKB1-liver kinase B1; AMPK-5′ AMP-activated protein kinase; P-phosphate; β-gly-β-glycerolphosphate; AA-ascorbic acid; Dex-dexamethasone (Image made with Biorender).

**Table 1 ijms-24-13829-t001:** Media composition of the shortlisted conditions. Composition of supplements added to the mineralization-inducing media used in cell culture. Phosphate and Calcium concentrations of DMEM are not added in this table. All concentrations are in Vol%, mmol L^−1^, µmol L^−1^, or nmol L^−1^ as described.

	CM 1	CM 2	CM 3	CM 4	CM 5	CM 6
FCS [Vol%]	2.5	2.5	5	5	5	
NaH_2_PO_4_ [mmol L^−1^]	0.95		0.65			
Na_2_HPO_4_ [mmol L^−1^]	0.95		0.65			
CaCl_2_ [mmol L^−1^]		2.5	0.2			
L-ascorbic acid [µmol L^−1^]				75	75	75
β-glycerolphosphate [mmol L^−1^]				10	12.5	7.5
Dexamethasone [nmol L^−1^]				10	5	5

**Table 2 ijms-24-13829-t002:** Media composition. Composition of supplements added to the mineralization-inducing media used in cell culture. All concentrations are in Vol%, mmol L^−1^, µmol L^−1^, or nmol L^−1^ as described.

Experiment	DMEM	P/S [%]	Na_2_HPO_4_: NaH_2_PO_4_ (1:1) [mmol L^−1^] Final Conc.	CaCl_2_ [mmol L^−1^] Final Conc.	FCS [%]	β-Glycerol-Phosphate [mmol L^−1^]	Ascorbic Acid [µmol L^−1^]	Dexame-Thasone [nmol L^−1^]
Phosphate or calcium concentration dependency								
Control	+	1	0.9	1.8	2.5	–	–	–
CM (Phosphate)	+	1	1.3, 1.8, 2.3, 2.8, 3.3, 3.8, 4.3	1.8	2.5	–	–	–
CM (CaCl_2_)	+	1	0.9	2.3, 2.8, 3.3, 3.8, 4.3, 4.8, 5.3	2.5	–	–	–
Phosphate and calcium concentration dependency								
Control	+	1	0.9	1.8	2.5	–	–	–
CM (Phosphate & fixed CaCl_2_)	+	1	1.3, 1.6, 1.9, 2.2, 2.5, 2.8, 3.1	2	2.5	–	–	–
CM (Fixed phosphate & CaCl_2_)	+	1	1.1	2.1, 2.4, 2.7, 3.0, 3.3, 3.6, 3.9	2.5	–	–	–
FCS concentration dependency								
Control	+	1	0.9	1.8	1, 2.5, 5, 7.5, 10, 12.5, 15	–	–	–
CM	+	1	0.9	1.8	1, 2.5, 5, 7.5, 10, 12.5, 15	10	75	–
Ascorbic acid concentration dependency								
Control	+	1	0.9	1.8	5	–	–	–
CM	+	1	0.9	1.8	5	10	25, 50, 75, 100, 125, 250	–
CM with dexamethasone	+	1	0.9	1.8	5	10	25, 50, 75, 100, 125, 250	10
β-glycerolphosphate concentration dependency								
Control	+	1	0.9	1.8	5	–	–	–
CM	+	1	0.9	1.8	5	1.5, 2.5, 5, 7.5, 10, 12.5, 15	75	–
CM with dexamethasone	+	1	0.9	1.8	5	1.5, 2.5, 5, 7.5, 10, 12.5, 15	75	10

**Table 3 ijms-24-13829-t003:** List of primers used.

Gene	Nucleotide Sequence	
β-actin	Forward	5′-CAACGAGCGGTTCCGATG-3′
	Reverse	5′-GCCACAGGATTCCATACCCAA-3′
Bone sialoprotein (*Lbsp*)	Forward	5′-TGAGTGACAGCCGGGAGAAC-3′
	Reverse	5′-AAGAAAGTAGCGTGGCCGGT-3′
Osteopontin (*Opn*)	Forward	5′-CTTTCACTCCAATCGTCCCTAC-3′
	Reverse	5′-GCTCTCTTTGGAATGCTCAAGT-3′

## Data Availability

The data presented in this study are available on request from the corresponding author. The data are not publicly available due to privacy or ethical restrictions.

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
