# Peer review of "Development, Establishment, and Validation of a Model for the Mineralization of Periodontium Remodelling Cells: Cementoblasts"

_ijms, 2023, doi:10.3390/ijms241813829_

Round 1
Reviewer 1 Report
In this paper, the authors present in detail a standardized operating protocol (SOP) for the systematic calcification of cementoblasts. The results are useful for understanding pathophysiological changes in the calcification process of the periodontal tissue-cementum complex. It will shed light on an unknown field in clinical research, especially in periodontal regenerative therapy. With regard to detailed research, the reader may request the following additional explanations. Please respond to the following comments.
Inactivated FCS is considered cytotoxic due to complement activation. It is speculated that this may also affect the apoptotic outcome of the cells in this study. An explanation or discussion of how the FCS was processed in this experiment is needed.
The experimental results report that high concentrations of phosphoric acid induce cementoblast apoptosis in a concentration-dependent manner. However, is such a high concentration expected in a biological environment?
The experimental results compare the two types of stained areas with respect to the different mechanisms of the calcification induction method. However, can area comparisons demonstrate even differences in calcium concentration of induced calcification?
The authors show that high concentrations of phosphate promote calcification pathways. It would be desirable to explain where the apoptosis-inducing death receptors affect within this cascade.
Author Response
Response to reviewer 1 comments: Summary: Thank you very much for taking the time to review this manuscript. Please find the detailed responses below and the corresponding revisions in track changes in the re-submitted files. Yes Can be improved Must be improved Not applicable Does the introduction provide sufficient background and include all relevant references? (x) ( ) ( ) ( ) Are all the cited references relevant to the research? (x) ( ) ( ) ( ) Is the research design appropriate? (x) ( ) ( ) ( ) Are the methods adequately described? (x) ( ) ( ) ( ) Are the results clearly presented? (x) ( ) ( ) ( ) Are the conclusions supported by the results? (x) ( ) ( ) ( ) In this paper, the authors present in detail a standardized operating protocol (SOP) for the systematic calcification of cementoblasts. The results are useful for understanding pathophysiological changes in the calcification process of the periodontal tissue-cementum complex. It will shed light on an unknown field in clinical research, especially in periodontal regenerative therapy. With regard to detailed research, the reader may request the following additional explanations. Please respond to the following comments. Comment 1: Inactivated FCS is considered cytotoxic due to complement activation. It is speculated that this may also affect the apoptotic outcome of the cells in this study. An explanation or discussion of how the FCS was processed in this experiment is needed. Response 1: We thank the reviewer for this suggestion. We have added the explanation in ma-terials and methods as follows: Page 13 lines 472-474 ‘. Sterile filtrated bovine serum was ac-quired from Gibco. Additionally, before usage, the FCS was heat inactivated for 30 min at 56 °C in a water bath and filtered again under sterile conditions. The FCS was divided in aliquots and stored at -20°C before use.’ And in the discussion on page 11 lines 349-351 as: ‘The FCS was heat inactivated to decrease the titer of heat labile complement proteins [23]. Lower concentra-tions of FCS e.g. less than 1% have been reported to be cytotoxic and can lead to higher apopto-sis [24].’
Comment 2: The experimental results report that high concentrations of phosphoric acid induce cementoblast apoptosis in a concentration-dependent manner. However, is such a high concentration expected in a biological environment? Response 2: We thank the reviewer for this comment. Under disease conditions the saliva phosphate concentrations drastically go up. We have further explained this in the discussion on page 10 line 317-319 as: ‘Renal disease or dental decay show a positive correlation between serum and saliva phosphate levels. Under disease conditions, saliva phosphate concentrations may go up to 13.7 ± 4.4 mg/dl which corresponds to 4.4 ± 1.4 mM [17].’ Comment 3: The experimental results compare the two types of stained areas with respect to the different mechanisms of the calcification induction method. However, can area comparisons demonstrate even differences in calcium concentration of induced calcification? Response 3: We thank the reviewer for this observation. The two staining methods have been used to exemplify the expected results for staining from the use of this protocol. The comparison between the six calcification based on area serve only as indications of the expected differences which need to be supplemented with precise calcium measurements demonstrated in figure 1-3. Moreover, the readers can choose between the staining methods to choose which they would like to employ for their experiments based on the quality of staining expected and shown for these cells. This has been further explained in the discussion on page 12 line 395-398 as follows: ‘The degree of staining positively correlates with the degree of calcification i.e., the amount of incorporated calcium. However, staining is best employed as a visualization of calcification and calcium measurement should be used to quantify the precise cellular calcium concentration as shown in figure 1-3.’ Comment 4: The authors show that high concentrations of phosphate promote calcification pathways. It would be desirable to explain where the apoptosis-inducing death receptors affect within this cascade. Response 4: We thank the reviewer for this comment. We have incorporated the desired explanation in the discussion on page 12 lines 418-420 as follows: ‘Since calcification and apoptosis often go hand in hand, it is important to consider that p38 [39], JNK [39], ERK [40] and ROCK [41] activation promotes apoptosis.’

Reviewer 2 Report
The topic of the manuscript is to investigate the impact of differential phosphate and calcium concentrations, a combination of differential phosphate and calcium concentrations, differential osteogenic supplements at different incubation times, as well as media supplements such as fetal calf serum to optimise a standard protocol for studying calcification in cementoblasts.
The title and the abstract of the article are informative. The Introduction briefly presents the issue of chronic kidney disease and the role of calcification in cementoblasts. The section "Material and Methods" precisely describes the chosen study design. The sections "Results" and “Discussion” are interestingly written, however, the paragraph about the study limitations and the more recent references should be supplemented. The Conclusions seem to be the "take-home" messages but could be extended.
Some following points must be clarified/corrected for the further processing of this article.
Merits-related comments:
1. The Introduction should highlight more broadly the background of the study and existing knowledge gaps.
2. The objective of the study is somewhat chaotically formulated. It should be more clearly worded and avoid the phrase "in conclusion".
3. The study limitations should be supplemented at the end of the Discussion.
4. It is suggested to add more recent articles from 2021-2023 to the references in the Introduction and the Discussion, e.g. considering the signalling pathways of CKD or the impact of the systemic processes on the functioning of the dental cells (e.g. 10.1038/s41392-022-01036-5, 10.3390/metabo13040520).
5. The Conclusions should be extended to further potential directions of the study, the use of the results for clinical purposes.
Technical comments:
1. References should be described as follows:
1. Author 1, A.B.; Author 2, C.D. Title of the article. Abbreviated Journal Name Year, Volume, page range.
Author Response
Response to reviewer 2 comments: Summary: Thank you very much for taking the time to review this manuscript. Please find the detailed responses below and the corresponding revisions in track changes in the re-submitted files. WE have improved the introduction, references and conclusions as suggested. Yes Can be improved Must be improved Not applicable Does the introduction provide sufficient background and include all relevant references? ( ) ( ) (x) ( ) Are all the cited references relevant to the research? ( ) (x) ( ) ( ) Is the research design appropriate? (x) ( ) ( ) ( ) Are the methods adequately described? (x) ( ) ( ) ( ) Are the results clearly presented? (x) ( ) ( ) ( ) Are the conclusions supported by the results? ( ) (x) ( ) ( )
The topic of the manuscript is to investigate the impact of differential phosphate and calcium concentrations, a combination of differential phosphate and calcium concentrations, differential osteogenic supplements at different incubation times, as well as media supplements such as fetal calf serum to optimise a standard protocol for studying calcification in cementoblasts. The title and the abstract of the article are informative. The Introduction briefly presents the issue of chronic kidney disease and the role of calcification in cementoblasts. The section "Material and Methods" precisely describes the chosen study design. The sections "Results" and “Discussion” are interestingly written, however, the paragraph about the study limitations and the more recent references should be supplemented. The Conclusions seem to be the "take-home" messages but could be extended. Some following points must be clarified/corrected for the further processing of this article. Merits-related comments: Comment 1: The Introduction should highlight more broadly the background of the study and existing knowledge gaps. Response 1: We thank the reviewer for this comment. We have clarified this in the introduction by addition of the sentences on page 2 lines 53-57 as follows: ‘Recently, in vivo methods are being developed to study alveolar defects resulting from diseases such as CKD (37191192), highlighting the need to study the clinical implications of dental dysfunction resulting from
other systemic diseases. However, in vitro methods to study the mechanisms involved have not been established yet.’ And on page 2 lines 69-72 as: ‘However, these mechanisms have not been studied in detail in context of dentoalveolar defects due to a lack of established in vitro models mimicking CKD like conditions. This hinders the development and establishment of new therapeutic interventions.’ Comment 2: The objective of the study is somewhat chaotically formulated. It should be more clearly worded and avoid the phrase "in conclusion". Response 2: We thank the reviewer for this comment. We have rephrased it as follows on page 2 lines 83-87: ‘This study facilitates development of an in vitro model to assess new therapeutic interventions for dental disorders accompanying chronic diseases. The adoption of this well-established and validated protocol will contribute to the standardization of experimental procedures when studying calcification in cementoblasts, promoting comparability among diverse research findings.’ Comment 3: The study limitations should be supplemented at the end of the Discussion. Response 3: We thank the reviewer for this suggestion. The limitations of the study have been added as follows on page 13 lines 453-461: ‘This study demonstrates that different calcification inducing supplements trigger different calcification inducing mechanisms. This mandates prior indepth knowledge and decision about the research question and expected affected mechanisms. If no such decision is made beforehand, two different experiments need to be carried out to study the effect of mediators of dental calcification. Moreover, this study is based on supplementation of cementoblasts with physiologic concentrations of calcification inducers, but as an In vitro study it is limited in its inability to fully replicate the complex interactions and physiological conditions of living organisms. This may lead to results which do not accurately reflect in vivo conditions.’ Comment 4: It is suggested to add more recent articles from 2021-2023 to the references in the Introduction and the Discussion, e.g. considering the signalling pathways of CKD or the impact of the systemic processes on the functioning of the dental cells (e.g. 10.1038/s41392-022-01036-5, 10.3390/metabo13040520). Response 4: We thank the reviewer for this suggestion. More recent studies including 10.1038/s41392-022-01036-5, 10.3390/metabo13040520 have been added to the introduction and discussion. Comment 5: The Conclusions should be extended to further potential directions of the study, the use of the results for clinical purposes. Response 5: We thank the reviewer for this suggestion. We have expanded the conclusion on page 13 lines 449-451 as follows: ‘Application of this in vitro protocol can facilitate studies focused on therapeutic interventions for clinical practice. This would lead to improved diagnostics and treatment strategies for dental conditions related to chronic diseases.’ Technical comments:
Comment 6: References should be described as follows: 1. Author 1, A.B.; Author 2, C.D. Title of the article. Abbreviated Journal Name Year, Volume, page range.
Response 6: We thank the reviewer for this comment. We have changed the reference format according to the MDPI citation style from Endnote.
Additional comment: Please add scale bar for Figures 4A, 4C, 5A, and 5C.
We thank the editor for this comment. The scale bars have been added to the figures

Round 2
Reviewer 1 Report
It was confirmed that detailed explanations were added and appropriate modifications were made.